



# Measurement of Henry's law and liquid-phase loss rate constants of peroxypropionic nitric anhydride (PPN) in deionized water and in n-octanol

Kevin D. Easterbrook,[1] Mitchell A. Vona,[1] Kiana Nayebi-Astaneh,[1] Amanda M. Miller,[1] and Hans D. Osthoff[1]

5    [1] Department of Chemistry, University of Calgary, 2500 University Drive N.W., Calgary, Alberta, Canada T2N 1N4

*Correspondence to*: Hans D. Osthoff (hosthoff@ucalgary.ca)

10                                          for *Atmos. Chem. Phys.*





**Abstract**

The Henry's law solubility ($H_S$) and liquid-phase loss rate constants ($k_l$) of the tropospheric trace gas
constituents peroxyacetic nitric anhydride (PAN, $CH_3C(O)O_2NO_2$; commonly known as peroxyacetyl
nitrate) and peroxypropionic nitric anhydride (PPN, $C_2H_5C(O)O_2NO_2$; also known as peroxypropionyl
nitrate) in deionized (DI) water and of PPN in n-octanol were measured using a flow bubble apparatus at
temperatures between 5.0 °C and 25.0 °C. For PAN in DI water, the observed values for $H_{S,aq}$ are consistent
with literature, whereas the solubility of PPN in DI water is slightly lower than literature values, ranging
from $H_S^{CP}(PPN)_{aq} = (1.49\pm0.05)$ M atm$^{-1}$ at 25.0 °C to $H_S^{CP}(PPN)_{aq} = (7.01\pm0.25)$ M atm$^{-1}$ at 5.0 °C (stated
uncertainties are at the 1σ level). The data are best described by $\ln(H_S^{CP}(PAN)_{aq}/[M\ atm^{-1}]) = -(17.8\pm0.3)$
$+ (5620\pm85)/T$ and $\ln(H_S^{CP}(PPN)_{aq}/[M\ atm^{-1}]) = -(19.5\pm1.7) + (5955\pm480)/T$ where $T$ is in Kelvin. For n-
octanol, the PPN solubility ranges from $H_S^{CP}(PPN)_{oct} = (88\pm5)$ M atm$^{-1}$ at 25.0 °C to $H_S^{CP}{}_{oct} = (204\pm16)$ M
atm$^{-1}$ at 5.0 °C and is best described by $\ln(H_S^{CP}(PPN)_{oct}/[M\ atm^{-1}]) = -(6.92\pm0.75) + (3390\pm320)/T$. n-
Octanol-water partition coefficients ($K_{OW}$) for PPN were determined for the first time, ranging from 59±4
at 25.0 °C to 29±3 at 5.0 °C. Observed loss rate constants in DI water are consistent with recent literature
and larger than the thermal dissociation rates for both PAN and PPN, consistent with a hydrolysis
mechanism, whereas $k_l$ values in n-octanol are significantly smaller than gas-phase dissociation rate
constants, likely owing to a "cage effect" in the organic liquid. The results imply that uptake of either PAN
or PPN on cloud water and organic aerosol is negligible but that uptake of PPN may constitute an
overlooked source of peroxy radicals in organic aerosol.



## INTRODUCTION

The peroxycarboxylic nitric anhydrides (PANs; also referred to as peroxyacyl nitrates, molecular formula $RC(O)O_2NO_2$, R≠H) are important trace gas constituents of the troposphere (Roberts, 1990, 2007; Gaffney and Marley, 2021). Formed as by-products by the same photochemistry that produces ozone ($O_3$) and photochemical smog, the relative abundances of PANs allow insights into the dominant hydrocarbons

involved in the $O_3$-production process (Williams et al., 1997; Roberts et al., 2007). Their formation and fate affects the budgets of nitrogen oxides (i.e., $NO_x = NO+NO_2$) and the abundances of radicals such as the hydroxyl radical (OH), the hydroperoxyl radical ($HO_2$) and organic peroxy radicals ($RO_2$) and, ultimately, $O_3$ and aerosol production rates (Roberts, 1990, 2007). The phytotoxic PANs are prone to thermal decomposition in warm climates but are longer-lived in parts of the troposphere where temperatures are

lower. Since their rate constants for photolysis and reaction with OH are relatively small in the lower troposphere (Talukdar et al., 1995; Harwood et al., 2003), dissolution in cloud or fog droplets followed by hydrolysis (Roberts et al., 1996; McFadyen and Cape, 1999) or stomatal uptake (Sparks et al., 2003; Teklemariam and Sparks, 2004) then constitute potential loss pathways. Hence, knowledge of their temperature-dependent Henry's law (H) and loss rate constants ($k_l$) for both aqueous and non-polar liquid

phases is of interest.

There are several variants of H constants in use (Sander et al., 2022). From a thermodynamic point of view, the dimensionless Henry's law solubility constant ($H_S^{CC}$), defined as the ratio of liquid over gas-phase concentration, is preferred. Atmospheric chemists typically use Henry solubility in units of M atm$^{-1}$ ($H_S^{CP}$; for simplicity, referred to as H in the remainder of this manuscript), which is defined as the ratio of liquid-

phase concentration (in M) over the partial pressure (in atm) and is related to $H_S^{CC}$ by:

$$HRT = H_S^{CC} \qquad (1)$$

Here, T is the temperature in Kelvin, and R is the universal gas constant (0.08205 L atm mol$^{-1}$ K$^{-1}$). As summarized in Appendix B of Staudinger and Roberts (2001), the temperature dependence of H is related to the enthalpy and entropy of solvation, $\Delta H_{sol}^0$ and $\Delta S_{sol}^0$, and is often described using a Van't Hoff-type

equation, i.e.,

$$\ln(H_S^{CC}) = \frac{\Delta S_{sol}^0}{R} - \frac{\Delta H_{sol}^0}{RT}, \qquad (2)$$

or, if H (i.e., $H_S^{CP}$) is used:

$$\ln(H) = A_H - \frac{B_H}{T} + C_H \ln(T) \approx A_H - \frac{B_H}{T}. \qquad (3)$$

Here, $A_H$, $B_H$ and $C_H$ are simple curve-fitting parameters. Equation (3) is used to parameterize H constants

by the National Aeronautics and Space Administration Jet Propulsion Laboratory (NASA-JPL) evaluation





panel of chemical kinetics data for use in atmospheric studies (Burkholder et al., 2020). Because $\Delta S_{sol}^0$ and $\Delta H_{sol}^0$ (and hence $A_H$ and $B_H$) for the PANs are not significantly temperature-dependent in the range of interest to atmospheric chemists, the third term is usually omitted (i.e., $C_H$=0) (Burkholder et al., 2020).

For the most abundant of the PANs, peroxyacetic nitric anhydride (PAN; often referred to as peroxyacetyl
nitrate), $H$ and $k_l$ constants in deionized (DI) water have been measured over a range of temperatures pertinent to the lower troposphere in a handful of studies (Kames and Schurath, 1995; Kames et al., 1991; Holdren et al., 1984; Frenzel et al., 2000; Lee, 1984). Measurements below room temperature are of particular interest because fog droplets and clouds (which can act as sinks for PANs) are generally more prevalent under those conditions (Koracin et al., 2001). For the second most abundant of the PANs,
peroxypropionic nitric anhydride (PPN, often referred to as peroxypropionyl nitrate), in contrast, there has only been a single study measuring $H$ and $k_l$ constants in DI water (Kames and Schurath, 1995), though it is clear from these data that the uptake of PAN and PPN on cloud droplets is negligible. Nevertheless, significant enhancements in the PPN:PAN ratio has been observed in fog-processed air in New England (Roberts et al., 1996), rationalized at least in part by reactive uptake of the peroxyacetyl (PA) radical
(formed from thermal dissociation of PAN) on fog droplets that is faster than the reformation of PAN via PA + NO₂ (Villalta et al., 1996).

The Henry's law solubility and liquid-phase loss rate constants of PAN in n-octanol have been reported by Roberts (2005) to gain insight into this molecule's potential uptake on organic aerosol and non-polar stomatal media. Analogous data for PPN do not exist, though PPN likely exhibits greater partitioning to
organic phases than PAN because of its larger aliphatic side chain. Knowledge of the air-octanol and air-water partitioning constants allows the widely used octanol-water partition coefficient ($K_{OW}$) to be calculated:

$$K_{OW} = \frac{H_S^{CC}(\text{octanol})}{H_S^{CC}(\text{aq})} = \frac{H_S^{CP}(\text{octanol})}{H_S^{CP}(\text{aq})} \tag{4}$$

This constant provides a direct estimate of hydrophobicity and of partitioning of a molecule from water to
organic media such as lipids, waxes, and natural organic matter, including organic aerosol (Mackay and Parnis, 2020), but has not been previously determined for PPN.

In this work, we report the $H$ and $k_l$ constants for PAN and PPN in DI water to corroborate the data by Kames and Schurath (1995) and for PPN in n-octanol at temperatures between 5.0 °C and 25.0 °C. We calculate $K_{OW}$ values for PPN and estimate rates for its wet deposition and uptake rates on organic aerosol
in the atmosphere.



## METHODS

Aliquots of PAN (or PPN) in tridecane solution were synthesized from reaction of acidified peroxycarboxylic acid solutions with concentrated nitric acid (Mielke and Osthoff, 2012; Williams et al., 2000) and stored in a freezer until use.

Gas streams containing PAN or PPN in nitrogen ($N_2$) were generated using diffusion sources as described by Furgeson et al. (2011) and depicted within the dashed box in Figure 1. At the start of each experiment, a solution containing PAN (or PPN) was placed in a 3-neck glass vessel submerged into a dewar containing an ice-water bath. The $N_2$ gas flow was set to ~40 standard cubic centimeters per minute (sccm) via a 50 μm diameter critical orifice (Lenox Laser) and a regulator back pressure of 25 pounds per square inch (psi, 1.7

bar). The source of the $N_2$ gas was the "blow-off" from a liquid $N_2$ dewar (not shown), chosen because of its high purity. The output of the diffusion source was directed to a fume hood via a 2-way perfluoroalkoxy (PFA) Teflon valve (Swagelok) and allowed to stabilize over a period of several hours.

Henry's law solubility and liquid-phase loss rate constants were measured using a jacketed, Pyrex bubble column apparatus (internal volume ~175 mL, diameter ~4.5 cm, height ~35 cm; Figure 1) similar to those

described by Kames and Schurath (1995) and Roberts (2005). The gas-phase PAN (or PPN) concentration downstream of the bubble column was monitored by a gas chromatograph with electron capture detector (GC-ECD) which sampled at an inlet flow rate of 75 sccm. The Varian GC-ECD described by Tokarek et al. (2014) was used for most of the experiments (see schedule in Tables S1-S4 in the supporting information (S.I.) section), though some measurements were also carried out using the Hewlett Packard (HP) GC

described in (Rider et al., 2015; Zborowska et al., 2021). The Varian GC-ECD was equipped with a 30 m long megabore column (Restek RTX-1701, film thickness 1 μm) and a 0.5 mL stainless steel sample loop (VICI Cheminert). It was operated at an oven temperature of 25 °C, an ECD temperature of ~100 °C, with $N_2$ carrier gas (from the aforementioned liquid $N_2$ dewar) at a carrier gas flow rate of ~40 mL min$^{-1}$, and a make-up gas flow rate of ~8.1 mL min$^{-1}$. The HP GC-ECD was equipped with a 15 m long megabore

column (Restek RTX-200, film thickness 1 μm) and a 2.0 mL stainless steel sample loop (VICI Cheminert). It was also operated at an oven temperature of at 25 °C and at an ECD temperature of 105 °C, but with helium (He) carrier gas at a flow rate of ~19.4 mL min$^{-1}$ and $N_2$ make-up gas at a flow rate of ~44.0 mL min$^{-1}$. The GCs were set to automatically inject every 120 s for experiments with PAN or every 360 s for experiments with PPN. For experiments with PAN, connecting tubing and fittings were constructed from

fluorinated ethylene propylene (FEP) Teflon, and the GC sampled through a Teflon filter (Pall, 2 μm pore size) housed in a PFA Teflon in-line filter holder (Cole-Parmer RK-06103-13). For experiments with PPN, tubing downstream of the bubble column were replaced with 1/8" (~0.32 cm) outer diameter (o.d.) stainless





tubing and fittings (Swagelok), and the GC inlet filter and holder were removed because of memory effects with the Teflon tubing and fittings.

In a typical experiment, the bubble column apparatus (right-hand side of Figure 1) was filled with a known volume ($V_l$) of liquid solution, i.e., either DI water (18 MΩ cm$^{-1}$, Thermo Scientific Barnstead Nanopure Model D11931) or n-octanol (ACS reagent grade, ≥99% purity, used as received). This apparatus was temperature-controlled using an external chiller-circulator (Lauda Proline RP 1290 for experiments with the Varian GC-ECD, or VWR 13271-100 for experiments with the HP GC-ECD). Once the desired

temperature was achieved, the gas stream containing PAN (or PPN) was bubbled through the water (or n-octanol) by opening the 2-way valve (top of Figure 1) and closing the 2-way valve connected to the waste line. After ~ 5 min the incoming gas stream was directed towards a waste line (by opening the 2-way valve connected to it). Gas-phase PAN (or PPN) mixing ratios ($c_g$) ranged from ~10 ppbv to ~300 ppbv at this stage. For experiments with PPN, the tee fitting was briefly (~30 s) back-flushed with $N_2$ delivered via a

calibrated mass flow controller (MFC, MKS Instruments, 500 sccm capacity for experiments with the Varian GC-ECD, 5000 sccm capacity for experiments with the HP GC-ECD) until the 2-way valve connecting the source to the bubble column apparatus and $N_2$ flow was closed. For experiments with DI water, the $N_2$ flow was humidified using a bubbler to minimize evaporation of solvent in the main bubble column. The PAN (or PPN) concentration decay was then monitored as a function of time ($t$) by either GC.

The experiment was repeated, systematically varying the (volumetric) flow rate to volume ratio ($\frac{\Phi}{V_l}$) as summarized in Tables S1-S4.

The concentration decays due to a combination of first-order loss and gas-liquid equilibration, i.e., partitioning of PAN (or PPN) from the liquid reservoir to the gas stream. Kames and Schurath (1995) showed that under these conditions, the analyte's concentration decreases according to the following

relationship:

$$-\frac{d}{dt}c_{g,t} = \left(\frac{\Phi}{H_S^{CC} V_l} + k_1\right)c_{g,t} \tag{5}$$

Here, $c_{g,t}$ is the analyte's concentration in the gas-phase at time $t$, $k_1$ is the loss rate constant of PAN (or PPN) in the liquid phase, and $H_S^{CC}$ is the dimensionless Henry solubility. Rearrangement of Eq. (5) yields:

$$\ln\left(\frac{c_{g,0}}{c_{g,t}}\right) = \left(\frac{\Phi}{H_S^{CC} V_l} + k_1\right)t \tag{6}$$

Since the chromatographic peak area ($A$) is proportional to $c_g$ (Tokarek et al., 2014), linear regression analysis of plots of $\ln\left(\frac{A_0}{A_t}\right)$ versus $\frac{\Phi}{V_l}$ yields the values of $1/H_S^{CC}$ as the slope and $k_1$ as the intercept. In this





work, peak areas were determined by fitting parameters of a Gaussian expression to the observed peaks as described by Tokarek et al. (2014) using an automated macro in Igor Pro (Wavemetrics Inc.).

The $H$ and $k_1$ constants measured in this work allow constraints to be placed on the heterogeneous uptake

of the PANs. Under conditions of rapid gas/liquid equilibration, the fraction of a trace gas's concentration in the liquid phase ($[X]_l$) relative to its gas-phase concentration ($[X]_g$) is equal to

$$\frac{[X]_l}{[X]_g} = H_S^{CC} \times L \tag{7}$$

and its first-order rate constant with respect to heterogeneous processing ($k_{het}$) is given by (Schwartz, 2003):

$$k_{het} = k_1 \times H_S^{CC} \times L. \tag{8}$$

Here, $L$ is the ratio of the liquid volume divided by the total volume of an air parcel. The formulism of Eq. (7-8) can be applied to liquid water ($L_{aq}$ and $k_{1,aq}$) or organic aerosol ($L_{org}$ and $k_{1,org}$). Hence, Eq. (8) may be used to calculate the loss rate of a molecule with respect to wet deposition ($k_{wet}$) or organic aerosol uptake ($k_{org}$).

Another quantity that may be derived from knowledge of $H$ and $k_1$ constants is the reactive uptake

probability ($\gamma$) for aqueous and (liquid) organic aerosol (Villalta et al., 1996):

$$\gamma \approx \frac{4 H_S^{CP} RT \sqrt{k_1 D_1}}{\omega} = \frac{4 H_S^{CC} \sqrt{k_1 D_1}}{\omega}. \tag{9}$$

Here, $\omega$ is the mean molecular speed ($= \sqrt{\frac{8RT}{\pi M}}$ where $M$ is the molecular weight, e.g., 0.1351 kg mol⁻¹ for PPN), and $D_1$ is the liquid phase diffusion coefficient (in m² s⁻¹), a quantity that can vary by many orders of magnitude depending on aerosol morphology (Shiraiwa et al., 2011). In this work, the magnitude of $D_1$ was

estimated using the semiempirical equation by Reddy and Doraiswamy (1967):

$$D_1 \approx \psi \frac{M_1^{1/2} T}{\mu V_1^{1/3} V_2^{1/3}}. \tag{10}$$

Here, $M_1$ is the molecular weight of the solvent, $\mu$ is the solvent viscosity (in mPa s), $V_1$ and $V_2$ are the molar volumes of the solute and solvent (in cm³ g⁻¹ mol), respectively, and $\psi$ is a constant equal to 8.5×10⁻⁸ when $V_2/V_1 > 1.5$ and 10×10⁻⁸ when $V_2/V_1 \leq 1.5$ (Reddy and Doraiswamy, 1967).

Once a value of $\gamma$ is known, the rate constant with respect to heterogeneous uptake in the kinetic limit (i.e., for submicron-sized aerosol) is then given by

$$k_{het} = \frac{1}{4} \omega \gamma S_A \tag{11}$$

where $S_A$ is the specific surface area density of the aerosol (Davidovits et al., 2006).



## RESULTS


Five consecutive chromatograms from a sample experiment are shown in Figure 2. In this example, the bubble column apparatus was filled with 150 mL of DI water at a temperature of 5.00 °C which was equilibrated with PPN. The dominant peak in the chromatogram is PPN, eluting at ~288 s. The peak eluting at ~87 s (whose area is ~5% that of PPN) is due to ethyl nitrate (EN), a by-product of the PPN synthesis

(Williams et al., 2000). There was also a small peak at ~136.5 s corresponding to PAN, another (though very minor) by-product of PPN synthesis. The relative elution times of EN, PAN, and PPN are consistent with their capacity factors of medium polarity columns (Roberts et al., 1989).

The chromatograms show a decrease in gas-phase concentrations over time as humidified $N_2$ is bubbled through the water column at a volumetric flow rate of 238 mL min⁻¹. Representative plots of $\ln\left(\frac{c_{g,0}}{c_{g,t}}\right)$ versus

$t$ for PPN in DI water are shown in Figure 3. The subset of the data plotted in Figure 2 are highlighted in bold and red colour. Example plots of $\ln\left(\frac{c_{g,0}}{c_{g,t}}\right)$ versus $t$ for PPN in n-octanol are shown in the S.I. as Figure S1. Plots such as those shown in Figures 3 and S1 were linear with Pearson correlation coefficients (r) > 0.99 for all experiments. Experiments with PPN in n-octanol usually required more time than those with DI water to equilibrate initially (as judged by initial deviation from linearity of plots of $\ln\left(\frac{c_{g,0}}{c_{g,t}}\right)$ versus $t$).

Because of the much slower decay rate, however, samples lasted for up to three days and allowed several flow conditions (i.e., different values of $\frac{\Phi}{V_l}$) to be probed in a single experiment.

Sample plots of $\frac{d}{dt}\ln\left(\frac{c_{g,0}}{c_{g,t}}\right)$ versus $\frac{\Phi}{V_l}$ (viz. Eq. (6)) for experiments with PAN and PPN in DI water at temperatures of 20 °C, 12.5 °C and 5.0 °C are shown in Figure 4. The error bars shown here and in the remaining Figures as well as the uncertainties stated in all Tables are at the 1σ precision level. Analogous

plots for experiments with PPN in n-octanol are shown in the S.I. as Figure S2. Linear regression analysis yielded slopes equal to $1/H_S^{CC}$ (Tables S5-S7), from which $H$ was calculated using Eq. (1). The results are summarized in Table 1, alongside available literature values.

Figure 5 graphically summarizes the $H$ constants measured in this work, along with available literature data. The data are presented as Van't Hoff plots, i.e., by plotting $H$ on a logarithmic scale versus $1000/T$ (with $T$

in Kelvin). Fits of Eq. (3) in Igor Pro to the data shown in Figure 5 yielded $C_H$ terms whose ±1σ uncertainty encompassed zero, such that the $C_H$ term was set to zero in subsequent fits. The results, i.e., values for $A_H$





and $B_H$ derived from Eq. (3) and linear fits to the data, are shown as solid lines in Figure 5 and are summarized in Table 2. Fits to literature values are shown as dashed lines.

Our measurements of $H$ for PAN in DI water are in good agreement with the majority of the available literature. Because of this, we combined our data with those of Kames and Schurath (1995) to yield the expression $\ln(H_S^{CP}(\text{PAN})_{\text{aq}}/[\text{M atm}^{-1}]) = -(17.8\pm0.3) + (5620\pm85)/T$ where $T$ is in Kelvin (Table 2). For PPN in DI water, on the other hand, our $H$ data are systematically lower than those by Kames and Schurath (1995) and are best described by the expression $\ln(H_S^{CP}(\text{PPN})_{\text{aq}}/[\text{M atm}^{-1}]) = -(19.5\pm1.7) + (5955\pm480)/T$ (Table 2).

Henry's law constants for PPN in n-octanol ranged from $(88\pm5)$ M atm$^{-1}$ at 25.0 °C to $(204\pm16)$ M atm$^{-1}$ at 5.0 °C and were larger than those of PAN in n-octanol (Figure 5). The data are best described by the expression $\ln(H_S^{CP}(\text{PPN})_{\text{oct}}/[\text{M atm}^{-1}]) = -(6.92\pm0.75) + (3390\pm320)/T$ (Table 2). The $H$ constants of both compounds exhibit similar temperature dependence with $\Delta H_{sol}^0$ (PAN)$_{\text{oct}} \approx -(33\pm2)$ kJ mol$^{-1}$ and $\Delta H_{sol}^0$(PPN)$_{\text{oct}} \approx -(28\pm3)$ kJ mol$^{-1}$ (Table 2).

Using the measured $H$ constants for PPN in DI water and n-octanol and Eq. 4, values of $K_{\text{OW}}$ of PPN were calculated. The octanol-water partition coefficient of PPN ranges from $59\pm4$ at 25 °C to $29\pm3$ at 5 °C, a factor of $\sim(4.1\pm0.6)$ larger than those reported by Roberts (2005) for PAN (Table 3).

Loss rate constants determined from the intercepts of plots such as those shown in Figure 4 are summarized in Table 4. Values for $\ln(k_l/[\text{s}^{-1}])$ are plotted versus $1000/T$ in Figure 6 alongside literature values, including data extracted from Figure 7 of Kames and Schurath (1995) and the rate constants for thermal dissociation of PAN and PPN (Kabir et al., 2014). All $k_l$ values in DI water are larger than (whereas all $k_l$ values in n-octanol are smaller than) the thermal decomposition rates of PAN and PPN (Figure 6). Table 5 summarizes linear regression parameters to the data shown in Figure 6, i.e., the temperature dependence of $k_l$. The loss rate constant of PPN in n-octanol increases when a small amount of a radical scavenger, α-tocopherol (Niki and Noguchi, 2004), is added to the solution (Figure S3) but is still smaller than PPN's thermal decomposition rate constant in the gas-phase as well as its loss rate constant in DI water (Table S8).

The results of this work allow constraints to be placed on the uptake of PPN by aqueous and organic matter in the atmosphere. The liquid water content (LWC or $L_{\text{aq}}$) of the atmosphere is highly variable and ranges from $<3\times10^{-6}$ g m$^{-3}$ in the cloud-free atmosphere under low aerosol loading (Nenes et al., 2021) to $>3$ g m$^{-3}$ in cumulonimbus clouds in the Tropics (Rosenfeld and Lensky, 1998). Using values of $k_{l,\text{aq}}$ and $H_{S,\text{aq}}^{\text{CC}}$ measured in this work and Eq. 8, we calculated the lifetimes with respect to hydrolysis ($\tau_{\text{wet}} = k_{\text{wet}}^{-1}$) of PAN and PPN for a range of atmospheric conditions including high (inorganic) aerosol loadings, clouds, and fog





at 293 K (Table S9) and 278 K (Table S10). The calculated wet deposition lifetimes of PPN (and PAN) vary from several days (e.g., in the presence of marine fog) to years.

Reactive uptake probabilities calculated using Eq. (9) are summarized in Table S11. For DI water, $\gamma$ values are on the order of $\sim 10^{-6}$ for both PAN and PPN at 278 K and 293 K. For n-octanol, $\gamma$ values are larger, varying between $4 \times 10^{-6}$ (at 278 K) and $2 \times 10^{-6}$ (293 K) for PAN and between $0.7 \times 10^{-5}$ (278 K) and $1.1 \times 10^{-5}$ (293 K) for PPN.

## DISCUSSION

**Solubility of PAN and PPN in DI water.** The $H$ data observed in this work for PAN (Figure 5, blue color) are in excellent agreement with the majority of the recent literature (Kames and Schurath, 1995; Kames et al., 1991; Frenzel et al., 2000; Lee, 1984) which gives confidence in the accuracy of the measurements presented in this manuscript. The only outlier is the data point by Holdren (1984) which is inconsistent with

all other measurements and appears to be in error. On the other hand, the $H$ constants of PPN in DI water observed in this work (Figure 5, red color) are systematically smaller than those by Kames and Schurath (1995), the only other study of water solubility of PPN. At 293.15 K, for example, a value of $H_{aq} = (2.67 \pm 0.18)$ M atm$^{-1}$ was observed in this work, $\sim 8\%$ lower than the literature value of $(2.90 \pm 0.06)$ M atm$^{-1}$. The gap widens at lower temperatures, e.g., to $\sim 17\%$ at 278.15 K. The reason (or reasons) for this difference

and whose measurement is ultimately more accurate are uncertain, though the results in this work are corroborated by the excellent agreement for PAN with literature (Figure 5).

The $k_l$ values for PAN and PPN in DI water (Figure 6) are also consistent with literature values: They are smaller than those reported for PAN by Kames et al. (1991) and marginally larger than the values reported for PAN and PPN by Kames and Schurath (1995). In DI water, all measured $k_l$ constants are larger than the

thermal dissociation rates of either PAN or PPN (shown as dashed lines in Figure 6), consistent with the hydrolysis mechanism hypothesized by Kames et al. (1991).

In general, accurate measurements of $k_l$ in unbuffered, DI water are challenging because the hydrolysis rate constants are larger in the presence of salts (Kames and Schurath, 1995) and pH dependent, i.e., base catalyzed (Lee, 1984) and slower in acidic solutions (Frenzel et al., 2000). Hence, impurities may

potentially bias experimental data for unbuffered, DI water. The largest source of impurities is the sample loading stage because gas streams containing PAN or PPN in high purity are challenging to generate. A contributing factor for the observed differences for PPN between this work and the study of Kames and Schurath (1995) may thus have been the method of PPN generation: In this work, a diffusion source was used, whereas Kames and Schurath (1995) used a photochemical source based on 3-pentanone photolysis





in the presence of NO$_2$ to generate PPN in situ. Both of these types of sources co-emit water-soluble impurities. It is clear that EN was emitted in this work, though only in minute quantities (Figure 2) that in all likelihood did not affect the solubility of PPN. In contrast, the photochemical source used by Kames and Schurath (1995) likely co-emitted, as they put it, "rather high mixing ratios" (~3000 ppmv) of 3-pentanone and would have also co-emitted a myriad of water-soluble photochemical by-products such as

hydroperoxides, aldehydes and α,β-dicarbonyls such as glyoxal (Rider et al., 2015). Furthermore, formation of photochemical side products would have been accentuated in their work since they utilized a 254 nm Hg lamp whose high operating temperature heightens undesired secondary photochemistry (Rider et al., 2015). It is unclear whether these impurities would have altered the bulk properties of the water contained in their bubble column and hence led to different $H$ and $k_1$ values, but it is certainly conceivable. Both diffusion and

photochemical sources may have also co-emitted some NO$_2$ generated from thermal decomposition of PAN or PPN, whose presence can increase the effective lifetime of PANs with respect to the thermal dissociation (Sehested et al., 1998), though fortunately this compound is many orders of magnitude less water-soluble than either PAN or PPN (Sander, 2015) such that the potential presence of NO$_2$ is inconsequential.

**Solubility of PAN and PPN in n-octanol and K$_{OW}$ values.** This work represents, to the best of our

knowledge, the first and only measurement of the solubility of PPN in an organic liquid and the first determination of K$_{OW}$ for this compound. The measured $H_{oct}$ and $K_{OW}$ constants are a factor of ~2.4±0.3 and ~4.1±0.6 larger for PPN than those reported by Roberts (2005) for PAN (Tables 1 and 3). These ratios are, qualitatively, rationalized by PPN's longer aliphatic side chain and are of similar relative magnitude as the $K_{OW}$ constants reported by MacKay et al (2006) for ethyl and methyl acetate (5.4:1.5 ≈ 3.6) as well as

ethyl and methyl formate (2.13:1.07 ≈ 2.0), i.e., the results are reasonable.

The $k_1$ values measured for PPN in n-octanol (Figure 6) are consistent with (but more precise than) those observed by Roberts (2005) for PAN and much smaller than the gas-phase thermal dissociation rate constants at all temperatures. The lower $k_1$ value is, qualitatively, rationalized by a solvent or "cage" effect, where PPN and PAN dissociate but the relative lipophilic fragments recombine faster than they react. When

a known radical scavenger, α-tocopherol, was added, $k_1$ increased but remained below the thermal dissociation limit (Figure S3 and Table S8), consistent with this interpretation.

**Atmospheric implications.** The results of this work allow constraints to be placed on the heterogeneous uptake of PPN in the atmosphere.

The calculated wet deposition lifetimes of PPN (and PAN) have units of years (Tables S9 and S10),

substantiating the conclusion by Kames and Schurath (1995) and others that wet deposition of these molecules in the troposphere is generally too slow to matter. The only potential exception is the marine environment where the phase ratio of $[X]_{aq}:[X]_g$ (Eq. (7); Tables S9 and S10) is large and can exceed $10^{-4}$





for both PAN and PPN and because Kames and Schurath (1995) measured a loss rate constant of $4.4\times10^{-3}$ s$^{-1}$ for filtered sea water, a factor of ~24 larger than that for DI water. The reason(s) for the larger $k_l$ values is

(are) not known but suggests the presence of one (or more) reaction pathway(s) in addition to hydrolysis. Assuming this larger $k_l$ value for marine boundary layer (MBL) fog or cloud droplets, $\tau_{wet}$ is on the order of tens of days for PPN and approximately half that for PAN (Tables S9 and S10). This suggests that wet deposition may potentially have contributed to the enhancement of the PPN:PAN ratio observed in the Gulf of Maine (Roberts et al., 1996). However, under the conditions of drizzling MBL fog as observed during

the New England Air Quality Study (NEAQS) in 2004, the suspended droplets would more likely resemble pure water droplets as opposed to droplets resembling oceanic surface water, which in turn would imply hydrolysis lifetimes on the order of hundreds of days and a negligible loss by wet deposition.

This work has shown the $H$ constants of PPN in n-octanol are larger than those for PAN in n-octanol, which suggests that PPN will be more soluble in organic aerosol (OA). Assuming $L_{org}=5\times10^{-11}$ (Roberts, 2005)

and further assuming that n-octanol is a good proxy of OA, we calculate (using Eq. (7)) an equilibrium fraction between $2\times10^{-7}$ at 5 °C and $1\times10^{-7}$ at 25 °C for the partitioning of PPN on OA, approximately double that of PAN but still orders of magnitude too small to be of consequence on the gas-phase concentration of PPN. For example, assuming an aerosol surface area characteristic of the polluted lower troposphere of $S_A = 1000$ μm$^2$ cm$^{-3}$ and 50% (liquid) organic aerosol content and using $\gamma = 1\times10^{-5}$ estimated

in this work for OA at 293 K (Table S11), the first-order rate constant of PPN uptake ($k_{org}$) would be ~$3\times10^{-7}$ s$^{-1}$, or a lifetime of ~43 days, which is negligible. Further, the viscosity of OA is often larger than assumed for the calculations in Table S11, i.e., $D_l$ can be many orders of magnitude smaller than $10^{-9}$ m$^2$ s$^{-1}$ (Shiraiwa et al., 2011). On the other hand, uptake of a radical precursor such as PPN on OA may very well initiate secondary radical chemistry within the particle phase that is believed to yield oligomeric and humic-

like substances (Lim et al., 2010), which in turn suggests that uptake of PPN should not be ignored. Our work provides the first estimates of $\gamma$ values for PPN (Table S11) since there are, to the best of our knowledge, no experimental measurements of $\gamma$ for PPN in the open literature. However, our $\gamma$ estimates for PAN are at odds with a recent publication by Sun et al. (2022) who reported $\gamma$ values in the range of (6 - 9)$\times10^{-5}$ for ambient aerosol at high relative humidity (RH) which are substantially larger than those

calculated for DI water in this work (~$10^{-6}$; Table S11). They argued that the large uptake on ambient aerosol would be driven by chemical loss of PAN within the particle phase via unspecified redox reactions, yielding a substantially larger $k_l$, which is reminiscent of the observation by Kames and Schurath (1995) for filtered sea water. However, because the uptake of PAN scales with $\sqrt{k_l}$, a factor of ~15 larger uptake in turn would require a ~200× larger liquid-phase reaction rate constant, which seems unrealistic or would

require some currently unknown chemistry.



**Data availability**

The $\frac{\mathrm{d}}{\mathrm{dt}}\ln\left(\frac{c_{\mathrm{g,o}}}{c_{\mathrm{g},t}}\right)$ and associated $T$ and $\frac{\Phi}{V_l}$ data for the experiments discussed are tabulated in the S.I. Raw data (i.e., digitized chromatograms) are available from the corresponding author (hosthoff@ucalgary.ca) upon request.


**Author contributions**

MAV (PPN in water using the Varian GC-ECD), KDE (PPN in n-octanol using the Varian GC-ECD), KNA (PAN in water using the HP GC-ECD), and AMM (PAN in water using the Varian GC-ECD) carried out the experiments and reduced the data. MAV and KNA synthesized PPN and PAN, respectively. HDO
conceptualized the experiments and drafted the manuscript with input from KDE.

**Competing interests**

The authors declare that they have no conflict of interest.

**Acknowledgments**

This work was made possible by the financial support of the Natural Sciences and Engineering Research Council of Canada (NSERC) in the form of a Discovery grant to HDO (RGPIN/03849-2016). KDE acknowledges an NSERC undergraduate student research award (USRA).





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



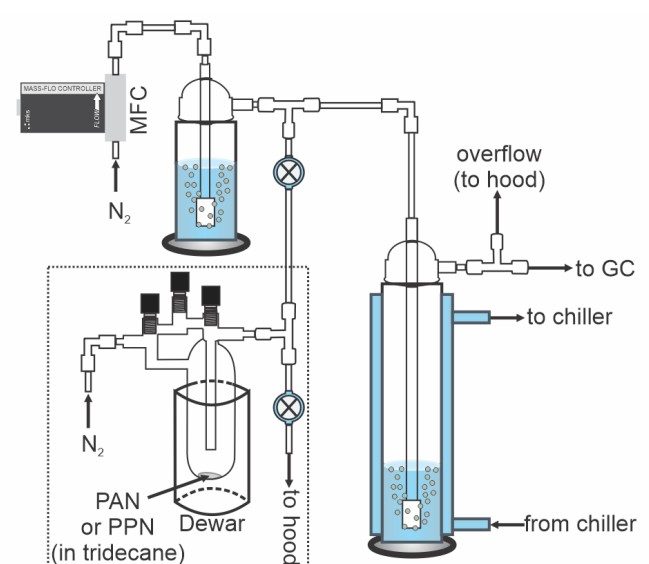

**Figure 1.** Schematic of the experimental apparatus (not to scale). The dashed line encompasses the PAN (or PPN) diffusion source; MFC = mass flow controller. The bubbler shown in the top left was removed for experiments with n-octanol.



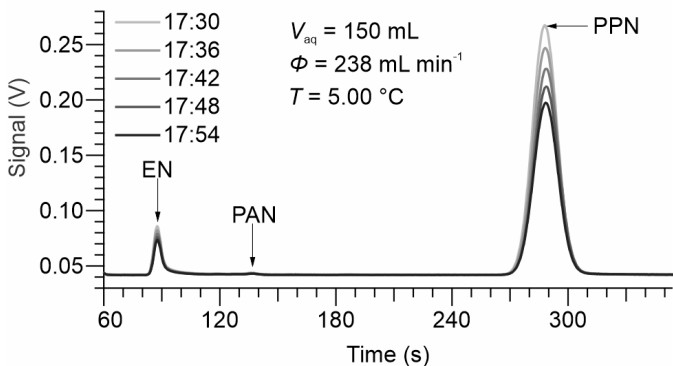

**Figure 2.** Sample chromatograms recorded by the Varian GC-ECD operated with $N_2$ carrier gas at a flow

rate of 40 mL min$^{-1}$. Here, the GC sampled the gas eluting from 150 mL DI water at a temperature of 5.00 ºC at a volumetric flow rate of 238 mL min$^{-1}$. The GC injected automatically every 360 s, at the time (in HH:MM) shown in the legend.





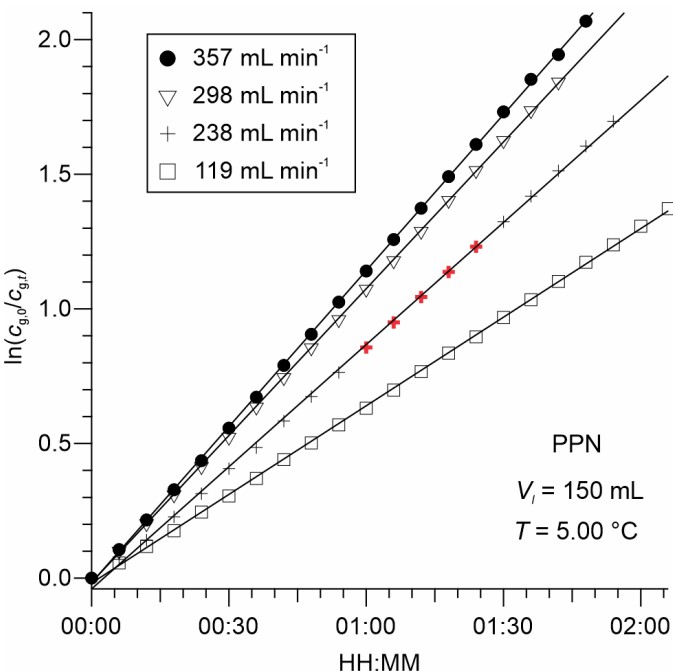

**Figure 3.** Plots of $\ln(c_{g,0}/c_{g,t})$ for PPN versus $t$, observed downstream from 150 mL of DI water at 5.00 °C for four different volumetric flow rates. The data from Figure 2 are highlighted in bold and red colour.

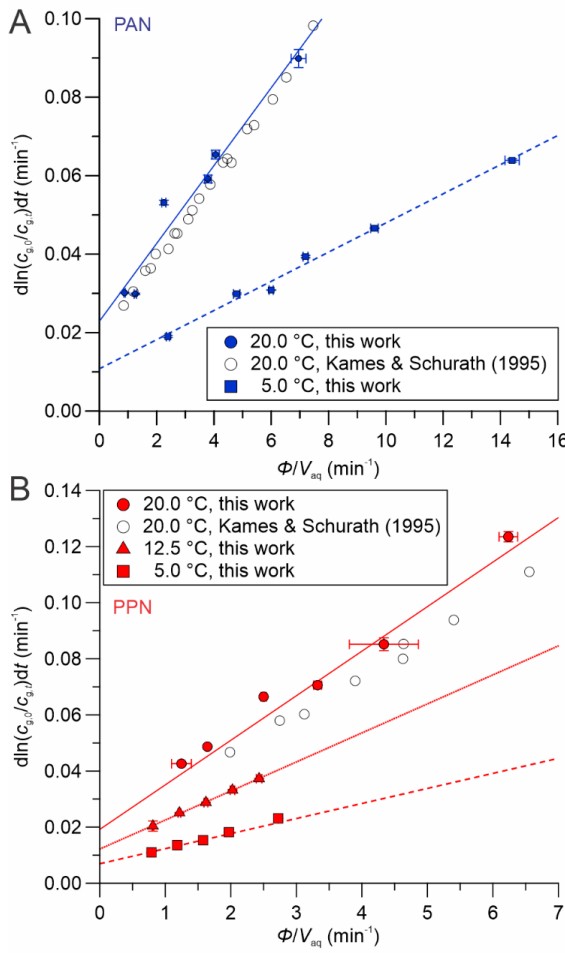

**Figure 4. (A)** Plots of $\mathrm{dln}(c_{g,0}/c_{g,t})/\mathrm{d}t$ versus $\Phi/V_l$ for PAN in DI water at 20.0 °C (●) and 5.0 °C (■).
Literature data at 20.0 °C were extracted from Figures 4 and 5 of Kames and Schurath (1995) using
"Engauge Digitizer" software and are shown as open symbols (○). The lines are linear fits to the data at
each temperature. **(B)** Same plot as panel (A) but for PPN in DI water and also showing data at 12.5 °C
(▲).



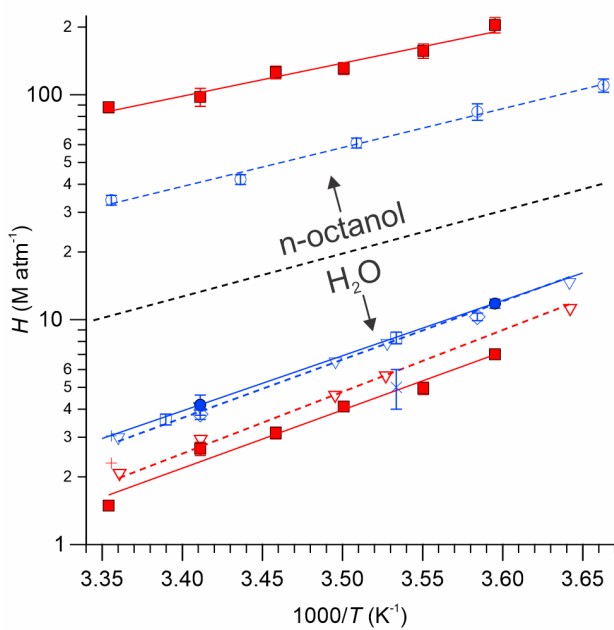

**Figure 5.** Henry's law constants for PAN (blue color) and PPN (red color) in n-octanol and DI water as
functions of temperature. Data from this work are shown as solid symbols (● for PAN and ■ for PPN). Data
from Figure 6 of Kames and Schurath (1995) for PAN and PPN were extracted using "Engauge Digitizer"
software and are shown as ▽ and --. Other literature data are shown as follows: □ (Lee, 1984). × (Holdren
et al., 1984). △ (Kames et al., 1991). ◇ (Frenzel et al., 2000). ○ and -- (Roberts, 2005). + (Raventos-Duran
et al., 2010). The dashed, black line serves as a visual indicator to separate results obtained with n-octanol
(top portion of graph) from those obtained with DI water (bottom).

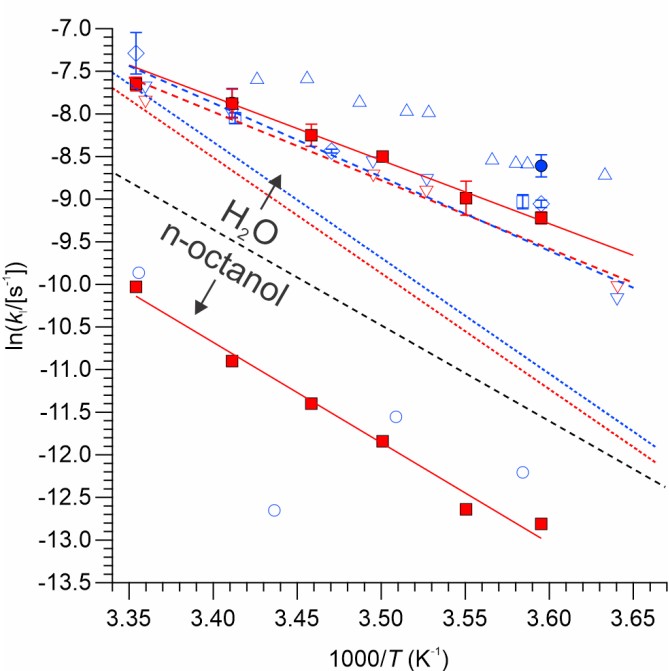

**Figure 6.** Natural logarithm of $k_1$ of PAN (blue color) and PPN (red color) in DI water and n-octanol plotted versus $1000/T$. Data from this work are shown as solid symbols (● for PAN and ■ for PPN). Literature data: × (Holdren et al., 1984). △ (Kames et al., 1991). ▽-- (Kames and Schurath, 1995). ◇ (Frenzel et al., 2000). ○ (Roberts, 2005). Error bars for the data by Roberts (2005) are omitted for clarity. The dashed blue and red lines (‴) are the expressions by Kabir et al. (2014) for thermal dissociation of PAN and PPN, respectively. The dashed, black line serves as a visual indicator to separate results obtained with n-octanol (bottom portion of graph) from those obtained with DI water (top).





**Table 1.** Summary of Henry's law solubility constants. N/A = not available (not tabulated). n/d = not determined.

| Compound and solvent | Reference | $H_S^{CP}$ (M atm$^{-1}$) | | | | | |
|---|---|---|---|---|---|---|---|
| | | 298.15 K | 293.15 K | 289.15 K | 285.65 K | 281.65 K | 278.15 K |
| PAN in DI water | (Lee, 1984) | n/d | 3.6±0.2 (295 K) | n/d | 8.3±0.5 (283 K) | n/d | n/d |
| | (Holdren et al., 1984) | n/d | n/d | n/d | 5±1 (283 K) | n/d | n/d |
| | (Kames et al., 1991) | N/A | 4.10±0.15 | N/A | N/A | N/A | N/A |
| | (Kames and Schurath, 1995) | N/A | 4.10±0.08 | N/A | N/A | N/A | N/A |
| | (Frenzel et al., 2000) | n/d | 3.8±0.2 | n/d | n/d | n/d | 10.3±0.4 (279 K) |
| | (Burkholder et al., 2020)[*] | 2.9[*] | 4.0[*] | 5.3[*] | 6.7[*] | 9.0[*] | 11.6[*] |
| | This work | n/d | 4.2±0.4 | n/d | n/d | n/d | 11.8±0.6 |
| PPN in DI water | (Kames and Schurath, 1995) | N/A | 2.90±0.06 | N/A | N/A | N/A | N/A |
| | This work | 1.49±0.05 | 2.67±0.18 | 3.14±0.19 | 4.11±0.07 | 4.96±0.31 | 7.01±0.25 |
| PAN in n-octanol | (Roberts, 2005) | 34±2 | 42±2 (291 K) | n/d | 61±3 (285 K) | n/d | 84±7 (279 K) |
| PPN in n-octanol | This work | 88±5 | 98±9 | 126±8 | 131±8 | 157±11 | 204±16 |

[*] Uncertainty factor of 2 to 10.





**Table 2.** Temperature dependence of Henry's law solubility constants derived by linear fitting of Eq. (3) with $C_H$=0 to the data shown in Figure 5. n/d = not disclosed. N/A = not applicable.

| Compound and solvent | Reference | $r^2$ (%) | $-B_H$ ($10^3$ K$^{-1}$) | $A_H$ (unitless) | $-R{\times}B_H = -\Delta H^0_{sol}$ (kJ mol$^{-1}$) | $R{\times}A_H$ (J K$^{-1}$ mol$^{-1}$) |
|---|---|---|---|---|---|---|
| PAN in DI water | (Lee, 1984)[*] | n/d | 5.9±0.6 | -19±2 | 49.2±4.7 | -157±16 |
| | (Kames et al., 1991) | n/d | 6.5±0.1 | -20.8±0.5 | 54.2±1.0 | -173.1±4 |
| | (Kames and Schurath, 1995) | n/d | 5.7±0.2 | -18.0±0.6 | 47.4±1.6 | -150±5 |
| | (Burkholder et al., 2020)[**] | N/A | 5.73 | -18.15 | 47.6 | -151 |
| | This work[***] | 99.89 | 5.62±0.09 | -17.8±0.3 | 46.7±0.7 | -148±3 |
| PPN in DI water | (Kames and Schurath, 1995) | n/d | 5.9±0.1 | -19.2±0.4 | 49.4±0.9 | -160±3 |
| | This work | 97.5 | 6.0±0.5 | -19.5±1.7 | 49.5±4.0 | -162±14 |
| PAN in n-octanol | (Roberts, 2005) | 99.1 | 4.0±0.2 | -9.9±0.8 | 33.1±1.8 | -82.1±6.3 |
| PPN in n-octanol | This work | 96.7 | 3.4±0.3 | -6.9±0.8 | 28.2±2.6 | -57.5±6.3 |

[*] Value for $\Delta S^0_{sol}$ and uncertainties were calculated by Kames and Schurath (1995) based on data provided by Lee (1984).

[**] Uncertainty factor of 2 to 10.

[***] Data from this work combined with those of (Kames and Schurath, 1995).





**Table 3**. n-Octanol - water partition coefficients of PAN and PPN calculated using Eq. (4). n/d = not determined.

| Compound | Reference | 298.15 K | 293.15 K | 289.15 K | 285.65 K | 281.65 K | 278.15 K |
|---|---|---|---|---|---|---|---|
| PAN | (Roberts, 2005) | 11.8±0.7 | 9.2±0.5 (291 K) | n/d | 8.8±0.5 (285 K) | n/d | 7.8±0.7 (279 K) |
| PPN | This work | 59.1±3.8 | 36.6±4.2 | 40.1±3.5 | 31.8±2.0 | 31.6±3.0 | 29.1±2.5 |






**Table 4.** Summary of liquid-phase loss rate constants. N/A = not available (not tabulated). n/d = not determined.

| Compound and solvent | Reference | $k_1$ ($10^{-5}$ s$^{-1}$) | | | | | |
|---|---|---|---|---|---|---|---|
| | | 298.15 K | 293.15 K | 289.15 K | 285.65 K | 281.65 K | 278.15 K |
| PAN in DI water | (Holdren et al., 1984) | n/d | 68±17 (298 K) | n/d | 21.7±0.5 (288 K) | n/d | 11.7±0.5 |
| | (Kames et al., 1991) | | 64±19 | N/A | N/A | N/A | N/A |
| | (Kames and Schurath, 1995) | | 34±1 | N/A | 19* | N/A | N/A |
| | (Frenzel et al., 2000) | | 32±2 | n/d | n/d | n/d | 12±1 (279 K) |
| | This work | | 38±7 | n/d | n/d | n/d | 18±2 |
| PPN in DI water | (Kames and Schurath, 1995) | N/A | 31±2 | N/A | 17* | N/A | N/A |
| | This work | 47±4 | 38±6 | 26±3 | 20.3±0.5 | 12±3 | 9.9±0.7 |
| PAN in n-octanol | (Roberts, 2005) | 5.2±1.9 | 0.32±0.25 (291 K) | n/d | 0.96±1.8 (285 K) | n/d | 0.5±2.6 (279 K) |
| PPN in n-octanol | This work | 4.4±0.2 | 1.8±0.3 | 1.1±0.2 | 0.7±0.1 | 0.3±0.2 | 0.3±0.2 |


* Value estimated from Fig. 7 of (Kames and Schurath, 1995).



**Table 5.** Temperature dependence of liquid-phase loss rate constants.

| Compound and solvent | Reference | $E_a$ (kJ mol$^{-1}$) | $A$ (s$^{-1}$) |
|---|---|---|---|
| PPN in DI water | (Kames and Schurath, 1995) | $54.5 \pm 3.1$ | $10^{6.2 \pm 0.6}$ |
| PPN in DI water | This work | $57.0 \pm 3.4$ | $10^{6.7 \pm 0.6}$ |
| PAN in n-octanol | (Roberts, 2005) | $67 \pm 56$ | $10^{7.1 \pm 10.2}$ |
| PPN on n-octanol | This work | $97.5 \pm 6.2$ | $10^{12.7 \pm 1.1}$ |