# Peer review of "Measurement of Henry's law and liquid-phase loss rate constants of peroxypropionic nitric anhydride (PPN) in deionized water and in n-octanol"

_Atmospheric Chemistry and Physics, 2022_

## Author Comment (AC1)

We thank both reviewers for their review of this paper and their insightful comments, which are reproduced in *black italic font* below. Our responses are shown in regular font. Revised text (as it appears in the revised manuscript) is shown in blue font, and removed text is . Line numbers are those of the revised manuscript with changes accepted.

*Reviewer 1: Review of ACP-2022-587, Easterbrook et al., Measurement of Henry's law and liquid-phase loss rate constants of peroxypropionic nitric anhydride (PPN) in deionized water and in n-octanol*

*This paper reports measurements of the solubility and first-order reaction rates of propionyl peroxynitrate (PPN) in water and n-octanol, and limited measurements of the solubility of acetyl peroxynitrate (PAN) in water. The data were then used to estimate atmospheric lifetimes of PPN against heterogeneous uptake in a number of scenarios. The measurements are solid and were well explained. I have only a few minor comments that need to be addressed and the paper should be acceptable for publication.*

**Response**: We thank the reviewer for this kind assessment.

*Line 45. It is not possible to make a blanket statement that the OH rate constants of PANs are low since MPAN (methacrylyl peroxynitrate) and APAN (acryloyl peroxynitrate) have unsaturated R- groups and are known or estimated to react rapidly with OH (Orlando and Tyndall, 2002; Orlando et al., 2002). The statement is true for PAN and PPN, but would benefit from having a range of lifetimes mentioned here.*

**Response**: We agree with the reviewer and have made the following changes, starting on line 45:

"For the two most abundant PANs, peroxyacetic and peroxypropanoic nitric anhydride (PAN and PPN, often referred to as peroxyacetyl and peroxypropionyl nitrate, respectively), the  rate constants for photolysis and reaction with OH are relatively small $(< 5 \times 10^{-7} \text{ s}^{-1})$ in the  troposphere (Talukdar et al., 1995; Harwood et al., 2003). Further, because the products of PAN (or PPN) photolysis may recombine and regenerate PAN (or PPN), their effective lifetimes are often much larger than suggested by these loss rate constants. For example, Harwood et al. (2003) calculated an effective lifetime of 4 months for PAN in the upper troposphere. Under these conditions, dissolution in cloud or fog droplets followed by hydrolysis (Roberts et al., 1996; McFadyen and Cape, 1999) or stomatal uptake (Sparks et al., 2003; Teklemariam and Sparks, 2004)  constitute potential loss pathways."

We also modified the text, starting on line 73:

"For the most abundant of the PANs,  PAN, ... For the second most abundant of the PANs,  PPN, in contrast, ... "

*Line 138. The mixing ratios noted here are quite high. Do they refer to the mixing ratios at the exit of the sources, or the entrance of the reactor. Most GC/ECD systems are limited in linearity at the high end. The technique used in this work relies on the detection scheme being linear with concentration over the range used in the experiment. How do you know your GC systems were linear at the highest mixing ratios used?*

*Line 155. The same idea as above, you need to justify the assumption of linearity.*

**Response**: We observed concentration decays over many orders of magnitude, all the way to the instrument's limit of quantification of ~10 pptv (Tokarek et al., 2014), without observing a change in d/dt ln($c_{g,0}/c_{g,t}$) - see, for example, Figure S1. From this, we conclude that the response was linear over the entire concentration range.

Because we already stated on line 198 "Plots such as those shown in Figures 3 and S1 were linear with Pearson correlation coefficients (r) > 0.99 for all experiments.", no changes were made in response to this comment.

*Line 180. Instead of the word "known" here, the word "estimated" seems more appropriate.*

**Response**: We agree and have changed the text as suggested: "Once a value of γ is estimated, the rate constant ..."

*Line 189. Was it possible to estimate H and k for ethyl nitrate? It looks like there is plenty of signal/noise for that peak.*

**Response**: Yes, it was. We are planning to publish these data in a future manuscript.

*Line 208 and Figure 5. The authors show, but do not explain points from Raventos-Duran et al., which appear to be estimates from structure-additivity relationships? This needs to be explained in the text.*

**Response**: We apologize and agree this should have been stated. We have expanded the text on line 209: "Figure 5 graphically summarizes the *H* constants measured in this work, along with available literature data, including estimates from the GROup contribution Method for Henry's law Estimate (GROMHE) structure-activity relationship model (Raventos-Duran et al., 2010). "

*Line 244. The Tables show lifetimes of 5 and 9 days, which are more than "several days"*

**Response**: The Merriam-Webster's dictionary definition of 'several' is 'more than two but fewer than many', with 'many' being defined as a 'a large number (of people or things)'. Further, Microsoft's word built-in thesaurus suggests 'quite a few' and 'numerous' as alternatives to 'several'.

We therefore believe that the word 'several' adequately describes the situation and have not altered the manuscript in response to this comment.

*Line 259. It would be useful to have the Kames and Schurath number for 278.15K in the table so we can see for ourselves.*

**Response**: We thank the reviewer for suggesting this. Kames and Schurath (2005) did not report $H_S^{CP}$ at 278.15 K; hence there is no entry in Table 1 at that temperature. However, they reported values at nearby temperatures and provided enthalpies and entropies of solvation (i.e., fitted their data); we calculated $H_S^{CP}$ values (e.g., 8.3 M atm$^{-1}$ at 278.15 K) based on their curve fit; the value of ~17% is based on a comparison of that curve-fit derived value with our experimental value at that temperature (7.0 M atm$^{-1}$) and was calculated by taking the difference and dividing by the average of both values.

Because we observe a difference of 17% when we compare our curve fit-derived values with those reported by Kames and Schurath (2005) across all temperatures investigated, we have reworded the sentence in question on line 262 as follows:

"The gap ~~widens at lower temperatures, e.g., to ~17% at 278.15 K~~ is larger (~17%) when $H_S^{cp}$ values are calculated based on the parameters in Table 2."

*Line 299. I think the authors mean "lipophobic" not "lipophilic". Lipophilic fragments would tend to react with n-octanol.*

**Response**: We thank the reviewer for catching this rather unfortunate typo and have changed the manuscript on line 309 as suggested: "... the relative lipophobic fragments ..."

*Lines 312-314. There is a difference in PAN and PPN lifetimes against uptake in marine fogs. However, the difference in thermal decomposition rates will be a bigger effect because of the shorter timescales over which thermal decomposition happens.*

**Response**: We agree with the reviewer but would like to point out that the net thermal loss is complicated in that environment because of recombination of the decomposition products, enhanced by the presence of some $NO_2$ from continental outflow. Because we ultimately conclude that uptake on marine boundary layer fog was negligible anyways, we have decided not to make changes to the text in response to the reviewer's comment.

*Table 1. The numbers from Burkholder et al are based on the Kames and Schurath work, so are not independent. Moreover, the uncertainty given by Burkholder et al seems inappropriately high. This may be due to the custom in chemical kinetics evaluations of assigning extra uncertainty when there is only one measurement reported in the literature.*

**Response**: Indeed, this is precisely what motivated this work - the large uncertainty in the NASA-JPL evaluation! Hopefully, they will be able to include our data in the next compilation and reduce those error bars. We modified the footnote in Table 1 (line 537) as follows: "[*] Literature evaluation with uncertainty factor of 2 to 10."

*References*

*Orlando, J. J. and Tyndall, G. S.: Mechanisms for the reactions of OH with two unsaturated aldehydes: Crotonaldehyde and acrolein, J. Phys. Chem. A, 106, 12252-12259, 2002.*

*Orlando, J. J., Tyndall, G. S., Bertman, S. B., Chen, W., and Burkholder, J. B.: Rate coefficient for the reaction of OH with CH2=C(CH3)C(O)OONO2 (MPAN), Atmos. Environ., 36, 1895-1900, 2002.*

*Reviewer 2 (Rolf Sander)*

*Easterbrook et al. measured Henry's law constants and liquid-phase loss rate constants of PAN and PPN. The manuscript is well-written, and I recommend publication in ACP after several minor issues have been dealt with.*

**Response**: We thank Dr. Sander for taking the time to review this manuscript and the many constructive comments below.

*- I think the introduction is too long. In particular, I don't think it is necessary to discuss the enthalpy and entropy of solvation, as they aren't even used in the rest of the manuscript. It should be sufficient to say that the parameters A_H, B_H and C_H are used to describe the temperature dependence.*

**Response**: We thank the reviewer for making this suggestion. The journal's author information web site does not provide any guidelines as to the length or composition of an introduction, and the particular paragraph in question is short (lines 61-72). In our view, it is necessary to mention enthalpy and entropy to justify the functional form of equation (3). Though we agree that this information is 'old news' for someone intimately familiar with Henry's law constants, it may not necessarily be familiar to the average reader of ACP. We have therefore decided not make changes to the text in response to this comment.

*- Please specify what temperature and pressure you are referring to when you use "sccm". There are many so-called "Standard" values, see https://en.wikipedia.org/wiki/Standard_temperature_and_pressure and https://en.wikipedia.org/wiki/Standard_cubic_centimetres_per_minute*

**Response**: We thank the reviewer for raising this point and have modified the text on line 106 as follows: "... ~40 standard (0 ˚C, 1 bar) cubic centimeters per minute (sccm) ..."

*- Line 125: FEP should not be called "Fluorinated ethylene propylene teflon" because it is not teflon.*

**Response**: We are uncertain if the reviewer raises a valid point here because there exists commercially available FEP tubing that is marketed under the Teflon brand name - see, for example https://ca.vwr.com/store/product/en/4637937/nalgene-890-teflon-fep-tubing-thermo-scientific or https://www.teflon.com/en/products/resins/fep-resins.

The specific tubing used in our experiments was purchased from Saint-Gobain Performance Plastics, who label their product "Chemfluor® Fluoropolymer Tubing", which in our view is functionally equivalent to the Teflon brand products.

In response to the reviewer's comment, we have modified the text on line 127 as follows:
"For experiments with PAN,  fluorinated ethylene propylene (FEP)  tubing (Saint-Gobain) and PFA Teflon fittings (Entegris) were used, and the GC sampled through a PFA Teflon filter (Pall, 2 μm pore size) housed in a PFA Teflon in-line filter holder (Cole-Parmer RK-06103-13) ... because of memory effects with the fluoropolymer tubing and fittings"

*- Line 138: The recommended symbol for gas-phase mixing ratios (mole fractions) is x, not c.*

**Response**: We thank the reviewer for raising this point. The International Union of Pure and Applied Chemistry (IUPAC) indeed recommends the symbol *x* for mole fractions (https://goldbook.iupac.org/terms/view/A00296). We have modified the text in question as follows:
"Gas-phase PAN (or PPN) mixing ratios  ranged from ..."
and now define the PAN (or PPN) concentration ($c_g$) on line 147.

*- Line 153: I would call the step from Eqn. (5) to Eqn. (6) an integration, not a rearrangement.*

**Response**: We have changed the manuscript as suggested: "Integration of Eq. (5) yields:"

*- Line 160: The quantity in Eqn. (7) should be called a "ratio", not a "fraction".*

**Response**: We agree. The sentence has been removed in response to the comment below.

*- Eqn (7): The factor "L" in this equation makes sense only if [X]_l and [X]_g are defined in a suitable way. Most likely, [X]_l refers to the volume of the air parcel, not to the liquid volume. Please provide the definitions of [X]_l and [X]_g.*

**Response**: We thank the reviewer for raising this point. Because [X]_l and [X]_g were not needed for any calculations, we have removed Eq. (7) and renumbered subsequent equations accordingly. The sentence on line 167 now reads: "Under conditions of rapid gas/liquid equilibration, the

$$\frac{[X]_l}{[X]_g} = H_S^{cc} \times L \tag{7}$$

 first-order rate constant with respect to heterogeneous processing ($k_{het}$) is given by (Schwartz, 2003):

$$k_{\text{het}} = k_l \times H_S^{cc} \times L. \tag{$\cancel{8}$7}$$

Here, $L$ is the ratio of the liquid volume divided by the total volume of an air parcel."

We also modified the text on line 317: "... the marine environment where the phase ratio  ($H_S^{cc}(\text{PAN}) \times L$; Tables S9 and S10) is large ..." and no longer refer to Eq. (7) on line 330.

*- Line 238: Why is a different definition of the liquid water contents used here than in Eqn. (7)? For consistency, I suggest to use only one version of the liquid water contents.*

**Response**: We have removed the duplicate definition (as $L$ was already defined):
"The liquid water content  of the atmosphere is highly variable and ..."

*- Line 304: I suggest to replace "have units of years" by "are on the order of years". The unit chosen for a quantity doesn't say anything about its magnitude.*

**Response**: We have modified the text as suggested: "The calculated wet deposition lifetimes of PPN (and PAN) are on the order of years "

*- Line 318: Change "This work has shown the H constants..." to "This work has shown _that_ H constants..."*

**Response**: Thank you for noticing this grammatical error - it has been corrected: "This work has shown that the *H* constants of PPN... "

*- Data availability: ACP requests depositing data in reliable public repositories. I don't think that providing data upon request is sufficient:*

*"Copernicus Publications requests depositing data that correspond to journal articles in reliable (public) data repositories, assigning digital object identifiers, and properly citing data sets..." (https://www.atmospheric-chemistry-and-physics.net/policies/data_policy.html)*

**Response**: In addition to the data already included in the SI, we have placed our raw and processed data to the University of Calgary's data repository (https://borealisdata.ca/dataverse/calgary) - the data are located at https://doi.org/10.5683/SP3/IYXA3G. The data availability section has been reworded to

"The $\frac{\mathrm{d}}{\mathrm{dt}} \ln\left(\frac{c_{\mathrm{g},0}}{c_{\mathrm{g},t}}\right)$ and associated $T$ and $\frac{\Phi}{V_l}$ data for the experiments discussed are tabulated in the S.I. Raw data (i.e., digitized chromatograms) have been placed in an online repository, accessible at https://doi.org/10.5683/SP3/IYXA3G, and are also available from the corresponding author (hosthoff@ucalgary.ca) upon request."

*- Figs. 5 and 6: If your plot program allows it, it would be nice to add a second x-axis at the top of the plot, showing T in addition to 1000/T.*

**Response**: Done.

*- Tab 1: It should be mentioned that, unlike the other entries in this table, Burkholder et al. (2020) is a literature review.*

**Response**: We modified the footnote in Table 1 (line 530) as follows:

"*Literature evaluation with u~U~ncertainty factor of 2 to 10."

*- Tab 1: Since calculated data from Raventos-Duran et al. are already shown in Fig. 5, I think that they should also be mentioned in Tab. 1.*

**Response**: The values have been added to Table 1 as requested, along with a footnote indicating that the Raventos-Duran data are based on SAR.

*Note that there is another publication with calculated Henry's law constants, including PAN and PPN: 10.5194/acp-17-7529-2017*

**Response**: We thank the reviewer for bringing the Wang et al. (*Atmos. Chem. Phys.*, 17, 7529-7540, 10.5194/acp-17-7529-2017, 2017) paper to our attention. Wang et al. estimated $K_{cc}$ constants (from which Henry's law constants in M/atm may be calculated) at 15 ˚C "using three alternative methods for direct gas–particle partitioning prediction: poly-parameter linear free-energy relationships (ppLFERs), the online calculator of SPARC Performs Automated Reasoning in Chemistry (SPARC), and the quantum-chemistry based program COSMOtherm)". Below is a summary of their results (taken from their paper's SI):

| $H_{S,aq}^{cc}$ | COSMOtherm | ppLFERs | SPARC | This work (289 K) |
|---|---|---|---|---|
| PAN | 1.8 | $1.2 \times 10^4$ | $2.5 \times 10^4$ | n/d |
| PPN | 0.87 | $9.3 \times 10^3$ | $9.8 \times 10^3$ | 74.5±4.5 |

| $H_S^{cp}$ | COSMOtherm | ppLFERs | SPARC | This work (289 K) |
|---|---|---|---|---|
| PAN | 0.073 M/atm | 480 M/atm | 1030 M/atm | n/d |
| PPN | 0.035 M/atm | 382 M/atm | 400 M/atm | 3.14±0.19 M/atm |

Wang et al. also estimated the equilibrium partitioning coefficients between a water-insoluble organic matter phase (WIOM) and the gas phase, $K_{WIOM/G} = C_{WIOM}/C_G$, which is equivalent to $H_{S,oct}^{cc}$ in this work:

| $K_{WIOM/G}$ | COSMOtherm | ppLFERs | SPARC | This work |
|---|---|---|---|---|
| PAN | $1.5 \times 10^3$ | $4.0 \times 10^3$ | $1.0 \times 10^5$ | n/d |
| PPN | $2.6 \times 10^3$ | $1.0 \times 10^3$ | $1.5 \times 10^5$ | $(3.0 \pm 0.2) \times 10^3$ |

For water-air partitioning, the ppLFERs and SPARC values calculated by Wang et al. are orders of magnitude larger than all other values listed in Table 1 and would (literally) be off scale if included in Figure 1. The COSMOtherm values, on the other hand, are orders of magnitude less than all other values (and also off-scale).

For the organic-air partitioning, the COSMOtherm and ppLFERs values are of similar magnitude as our experimental value, whereas the SPARC value is a factor of 100× off.

Because the values reported by Wang et al. are so divergent, we have chosen not to include them in our Tables and Figures as we felt that a comparison of our experimental data with these divergent and hence uncertain theoretical predictions would not be appropriate. However, we now acknowledge their existence by inserting the following on line 300:

"Wang et al. (2017) predicted gas-water and gas-organic partitioning coefficients for PAN and PPN using three independent prediction methods; however, their predicted values diverged considerably between each other and differed by several orders of magnitude from the experimental values reported in this work, especially for air-water partitioning. This is broadly consistent with the conclusions of their paper that noted challenges of theoretical methods to accurately predict Henry's law constants, particularly for multifunctional compounds such as PAN or PPN."

*- It is nice to see that the authors are following the new IUPAC recommendations. However, a very minor comment is that lower case superscripts should be used for c and p because these are the symbols for concentration and pressure, respectively.*

**Response**: We thank Dr. Sander for this suggestion and have changed the CP and CC superscripts to lower case throughout the manuscript and the SI accordingly.

*- The Engauge Digitizer is not sorted alphabetically in the list of references.*

**Response**: We thank the reviewer for noticing. The Engauge Digitizer software was written by Markum Mitchell - Endnote inserted the reference in the correct alphabetical order (under M) but omitted the author for some odd reason. This has been fixed.